# A Proteomic Analysis for the Red Seaweed *Asparagopsis taxiformis*

**DOI:** 10.3390/biology12020167

**Published:** 2023-01-20

**Authors:** Zubaida P. Patwary, Min Zhao, Tianfang Wang, Nicholas A. Paul, Scott F. Cummins

**Affiliations:** 1Centre for Bioinnovation, University of the Sunshine Coast, Maroochydore, QLD 4558, Australia; 2School of Science, Technology and Engineering, University of the Sunshine Coast, Maroochydore, QLD 4558, Australia; 3Department of Aquaculture, Faculty of Fisheries, Hajee Mohammad Danesh Science and Technology University, Dinajpur 5200, Bangladesh

**Keywords:** *Asparagopsis taxiformis*, seaweed, proteomics, mass spectrometry, sporophyte, gametophyte

## Abstract

**Simple Summary:**

There is increasing interest in the red seaweed *Asparagopsis taxiformis* due to its potential use as a feed supplement to knock out methane emissions of ruminants. Yet, there is little understanding of its molecular biology, including the proteins that facilitate photosynthesis, metabolite biosynthesis, growth and reproduction. This study used different extraction procedures to elucidate proteins from *A. taxiformis* (Lineage 6) sporophytes (diploid stage) and gametophytes (haploid stage). The outcome is the most comprehensive overview of expressed proteins in a seaweed, with interesting insights into the potential functions of proteins. This provides a foundation for future targeted protein extraction from *A. taxiformis* in search of potential bioactive molecules relevant to health and ecology.

**Abstract:**

The red seaweed *Asparagopsis taxiformis* is a promising ruminant feed additive with anti-methanogenic properties that could contribute to global climate change solutions. Genomics has provided a strong foundation for in-depth molecular investigations, including proteomics. Here, we investigated the proteome of *A. taxiformis* (Lineage 6) in both sporophyte and gametophyte stages, using soluble and insoluble extraction methods. We identified 741 unique non-redundant proteins using a genome-derived database and 2007 using a transcriptome-derived database, which included numerous proteins predicted to be of fungal origin. We further investigated the genome-derived proteins to focus on seaweed-specific proteins. Ontology analysis indicated a relatively large proportion of ion-binding proteins (i.e., iron, zinc, manganese, potassium and copper), which may play a role in seaweed heavy metal tolerance. In addition, we identified 58 stress-related proteins (e.g., heat shock and vanadium-dependent haloperoxidases) and 44 photosynthesis-related proteins (e.g., phycobilisomes, photosystem I, photosystem II and ATPase), which were in general more abundantly identified from female gametophytes. Forty proteins were predicted to be secreted, including ten rhodophyte collagen-alpha-like proteins (RCAPs), which displayed overall high gene expression levels. These findings provide a comprehensive overview of expressed proteins in *A. taxiformis*, highlighting the potential for targeted protein extraction and functional characterisation for future biodiscovery.

## 1. Introduction

Seaweeds are integral components of marine ecosystems, providing fundamental ecosystem services (e.g., oxygenation, de-acidification and nutrient uptake), along with economic growth through their use as food (e.g., basic ingredients or as dietary supplements) [1,2]. Increased interest in seaweeds has led to expanding global seaweed aquaculture production, recently estimated at 30.1 million tonnes with an annual value of USD 11.7 billion [3]. Seaweeds are also emerging sources for natural products with potential use in, for example, biotechnology and pharmacology [4].

Recently, the red seaweed *Asparagopsis taxiformis* (order Bonnemaisoniales) has gained interest due to its potential to mitigate methane-associated climate change [5]. *A. taxiformis* supplementation (5% organic matter) in ruminant feed knocks out enteric methanogenesis by 99%, without significant negative impacts on volatile fatty acid profiles and organic matter digestibility [6]. Therefore, the *A. taxiformis* genome sequence is of interest as a tool to identify the basic gene components of its metabolism and biosynthesis [7]. However, the genome is also a resource for high-throughput proteomic investigation, which can be used to discover bioactive compounds along with their biosynthesis mechanisms, such as for medicinal plants with their health-promoting properties [8,9] and in translational research [10,11]. The integration of transcriptome data to eukaryotic organism proteomics is complementary, since the transcriptome avoids potential errors caused by introns and other non-coding sequences [12]. Moreover, transcriptome-informed protein databases are known to reduce the ambiguity of protein identifications due to the higher proportion of specific peptides and a correspondingly lower proportion of shared ones [13,14]. All of these applications benefit from increased sensitivity that comes with integrating proteomics with mass spectrometry. Mass spectrometry (MS)-based proteomics has become a remarkably powerful approach for large-scale high-throughput protein identification, helping to inform biological questions.

To date, there have only been a handful of proteomic studies of seaweeds. There are four proteomic studies with detailed protein extraction protocols in seaweeds followed by proteomic investigation [15]: *Gracilaria changii* [16], *Eucheuma cottonii* (now *Kappaphycus alvarezii*) [17], *Ecklonia kurome* [18], *Scytosiphon gracilis* and *Ectocarpus siliculosus* [19]. These studies demonstrated extraction methods of proteins, followed by two-dimensional electrophoresis (2-DE) [16,18,19] and SDS-PAGE separation [17]. One of the key limitations for expanding MS-based proteomics in seaweeds is sample preparation, and more specifically the extraction of proteins remains a bottleneck for obtaining high-quality MS data. The complex polysaccharides within seaweed cell walls can negatively impact protein purification and analysis [20], and each seaweed has different polysaccharides and therefore needs to have bespoke protein extraction methods. This is also a problem for commercial extraction of proteins from seaweed [21].

This study extends upon our proteomic investigation of the *A. taxiformis* diploid sporophyte [22] through the additional interrogation and comparisons of male and female haploid gametophytes and the use of different extraction procedures to partition soluble and insoluble proteins derived from the nuclear genome. The outcomes establish new insights into the molecular make-up of seaweed proteins, including signalling, photosynthesis and secretory proteins.

## 2. Materials and Methods

### 2.1. Sample Collection

Sporophytes and gametophytes (male and female) of the red seaweed *Asparagopsis taxiformis* were sampled randomly from subtidal areas at ~0.5–3 m depth by snorkel from Moffat Beach (26°47′35.88′′ S, 153°08′18.96′′ E) located in Caloundra on the Sunshine Coast, Queensland, Australia. Samples were transported in seawater to the laboratories at the University of the Sunshine Coast within 1 h of collection. In the laboratory, the samples were cleaned with sterile seawater to remove visible mud, epiphytes and epibionts. Samples were then frozen (in 1.5 mL Merck tubes) in liquid nitrogen and stored at −80 °C until use.

### 2.2. Reference Genome and Transcriptome-Derived Protein Databases

Based on our reference genome (NCBI BioProject PRJNA809757), we derived a protein database using an integrative bioinformatics approach [22]. To obtain a reference transcriptome-derived protein database, total RNA was extracted from three anatomically different parts of individual male and female gametophytes, namely the tip (taken ~1 cm in length from the apex), main axis (upright section void of lateral branchlets) and the rhizome. Sporophytes (filamentous pom-pom-like structure) were also sampled. Five biological replicates for each except four replicates for the female main axis were processed for total RNA isolation using the RNeasy Plant Mini Kit (Qiagen, Germany) and an additional DNase I (Thermo Fisher Scientific, Waltham, MA, USA) treatment to remove residual genomic DNA, both following the manufacturer’s instructions. The RNA quantity and quality were assessed using a NanoDrop 2000 (Thermo Fisher Scientific, Waltham, MA, USA) spectrophotometer and Agilent Bioanalyzer RNA integrity number (RIN). Whole mRNA-seq (paired-end) was performed at a depth of at least 3 Gb by Novogene (Republic of Singapore, Singapore) using the Next-Generation Illumina NovaSeq 6000 platform, according to the standardised Illumina pipeline.

In total, 111.9 GB of clean raw sequence data were generated and deposited into the GenBank database under accession number PRJNA869876. To provide comprehensive and non-redundant amino acid sequences, we adapted a computational pipeline by integrating de novo transcriptome assembly and genome-based proteome prediction. In brief, the high-quality trimmed reads were used to generate a total of 7 different tissue-based de novo transcriptome assemblies (female tip, female main axis, female rhizome, male tip, male main axis, male rhizome and sporophyte), each including the combined reads from all replicates using the RNA-seq tool in CLC Genomics Workbench (Ver. 21.0.3) with default parameters. To provide all the possible protein-coding sequences, the aforementioned assemblies were then translated to amino acid sequences by a combination of Orf Predictor (PMID: 15980561) and Augustus [16]. In our draft genome annotation, we predicted a proteome with 10,867 proteins [22]. The redundancy acquired from pooling the seven assemblies and genome-based proteome was reduced using CD-HIT (PMID: 22772836) by applying 90% identity with a parameter clustering threshold. The final non-redundant proteome consists of a total of 2,320,110 proteins in combined transcriptome assemblies which will be an important resource for studies in *A. taxiformis*, on its life history transitions, its culture and selective breeding.

### 2.3. Protein Extraction and Proteomic Analysis

An overview of the protein extraction approach is shown in Figure 1. Frozen *A. taxiformis* gametophyte (male and female) tips and whole sporophytes were manually powdered using a sterile mortar and pestle in liquid nitrogen. Proteins were extracted using two different solutions: (1) Tris-HCl, pH 7.4 buffer, and (2) 6 M urea, chaotropic agent, with and without acetone precipitation. For a single extraction, approximately 100 mg of powdered sample was placed into a 50 mL tube. To obtain the water-soluble proteins, the powdered sample was homogenised with 20 mM Tris-HCl pH 7.8 buffer containing protease inhibitor (100×, Merck), briefly vortexed and centrifuged at 4000× *g* for 25 min at 4 °C, followed by separate extraction for supernatant and pellet.

The initial supernatant was extracted using both acetone and non-acetone procedures. For non-acetone precipitation, the supernatant was collected and centrifuged through an Amicon Ultra-4 centrifugal filter unit (3 kDa; Millipore Corporation, Billerica, MA, USA) at 4000× *g* for 40 min (4 °C) to purify proteins, followed by lyophilisation in a vacuum concentrator (Express SpeedVac Concentrator SC250, Thermo Fisher Scientific, Waltham, MA, USA). This extraction resulted in two protein samples: one over 3 kDa (Tris “Upper”) and one below 3 kDa (Tris “Lower”). However, for acetone precipitation, the supernatant was mixed with a pre-chilled (−20 °C) solution of 100% acetone (volume of acetone 4-times that of the protein samples to be precipitated) and kept overnight at −20 °C for precipitation. The supernatant was then centrifuged at 4000× *g* for 25 min at 4 °C, decanted, and the protein pellets were air-dried to evaporate the acetone from the uncapped tube (Tris “Acetone”).

For the water-insoluble proteins, the remaining pellets were resuspended in lysis buffer (6 M urea), vortexed and centrifuged at 4000× *g* for 25 min (4 °C). As above, the supernatant was collected, centrifuged through an Amicon Ultra-4 centrifugal filter unit and lyophilized in a vacuum concentrator. This lysis buffer extraction resulted in two protein samples: one over 3 kDa and one below 3 kDa, labelled as urea “Upper” and urea “Lower”, respectively. Similarly, acetone precipitation of the supernatant was carried out as above, designated as the urea “Acetone” sample.

Lyophilised protein samples were separated by solid-phase extraction (SPE) using a C18 column, 500 mg sorbent per cartridge, 55–105 µm particle size (Waters), except for the Tris “Acetone”, urea “Acetone” and Tris “Upper” samples after being dissolved in 0.1% trifluoroacetic acid (TFA) in Milli-Q water. In brief, the columns were conditioned and equilibrated with 100% acetonitrile (ACN) and 0.1% trifluoroacetic acid (TFA) in Milli-Q water, respectively. Then, samples were loaded followed by washing two times with 0.1% TFA in Milli-Q water and finally eluted by 0.5% acetic acid in 70% can in Milli-Q. Afterwards, the SPE eluate was centrifuged at 4000× *g* for 8 min (4 °C), and evaporated in a vacuum concentrator (Express SpeedVac Concentrator SC250, Thermo Fisher Scientific, Waltham, MA, USA) and stored at −20 °C prior to further analysis.

In the next step, samples were resuspended in 6 M urea and the protein concentration was determined using UV spectrophotometry (NanoDrop ND-2000). The trypsin digestion of the samples except for Tris “Bottom” and urea “Bottom” were performed as described by [23]. Following trypsin digestion, ultra-high performance liquid chromatography (uHPLC) tandem mass spectrometry (MS) was performed to identify proteins. A 6 µL aliquot of each sample was injected onto a 50 mm × 300 µm C18 trap column (Agilent Technologies, Mulgrave, Australia) and de-salted for 5 min using 0.1% formic acid (aqueous) at 30 µL/min. The trap column was then placed in-line with a 150 mm × 75 µm 300SBC18, 3.5 µm nano HPLC column (Agilent Technologies, Mulgrave, Australia) for LC-MS analysis. Peptides were eluted at a flow rate of 300 nL/min using a linear gradient of 1–40% solvent B (solvent A = 0.1% formic acid in Milli-Q water; solvent B = acetonitrile: 0.1% formic acid 90:10) over 35 min followed by a steeper gradient from 40% to 80% solvent B over 5 min. The gradient was held at 80% solvent B for 5 min to wash the column and returned to 1% solvent B for equilibration prior to the next sample injection. The ionspray voltage was set to 2400 V, declustering potential (DP) 100 V, curtain gas flow 25, nebuliser gas 1 (GS1) 12 and interface heater at 150 °C. The mass spectrometer acquired 500 ms full-scan TOF-MS data followed by 20 × 50 ms full-scan product ion data in an Information-Dependent Acquisition (IDA) mode. Full-scan TOF-MS data were acquired over the mass range 350–1800 *m*/*z* and for product ion MS/MS 100–1800 *m*/*z*. Ions observed in the TOF-MS scan exceeding a threshold of 100 counts and a charge state of +2 to +5 were set to trigger the acquisition of product ion MS/MS spectra of the resultant 20 most intense ions. The data were acquired and processed using Analyst TF 1.5.1 software (ABSCIEX, Concord, Canada).

### 2.4. Proteomic Identification and Annotation

Proteins were analysed using PEAKS v7.0 (BSI, Toronto, Canada) against the protein database built from the reference *A. taxiformis* (L6) genome database (10,863 proteins) [22], as well as from the *A. taxiformis* de novo transcriptome database (2,320,110 proteins). De novo sequencing of proteins, database searches and characterisation of specific post-translational modifications were used to analyse the raw data; false discovery rate was set to ≤1%, and [−10*log(p)] was calculated accordingly, where p is the probability that an observed match is a random event. PEAKS parameters were the following: (i) precursor ion mass tolerance, 15 ppm; (ii) fragment ion mass tolerance, 0.1 Da (the error tolerance); (iii) tryptic enzyme specificity with three missed cleavages allowed; (iv) monoisotopic precursor mass and fragment ion mass; (v) a fixed modification of cysteine carbamidomethylation; and (vi) variable modifications including lysine acetylation, deamidation on asparagine and glutamine, oxidation of methionine and conversion of glutamic acid and glutamine to pyroglutamate. MS raw data were deposited into the PRIDE database under accession number PXD035669.

OmicsBox software v2.1.2 (BioBam Bioinformatics SL, Valencia, Spain) was used for protein functional annotation (*e*-value cut-off 10^−3^). Photosynthesis- and stress-related proteins were identified based on gene annotations from the red seaweed *Chondrus crispus* [24], while gene annotations of *A. taxiformis* (Lineage 6) were used to identify predicted secreted proteins [22].

### 2.5. Quantitative RNA-Seq Analysis

Trimmed RNA-seq data were annotated by alignment against the *A. taxiformis* genome using CLC Genomics Workbench (Ver. 21.0.3) with default parameters. The quantification of RNA-seq for each gene was converted to transcripts per kilobase million (TPM). A Venn diagram showing all the extracted proteins among sporophyte, male and female tips was prepared using Good Calculators. A heatmap of all proteins from sporophyte (n = 5), male (n = 5) and female (n = 5) tips were constructed utilising the Z-score of average TPM as input data, while two heatmaps for photosynthesis-related and stress-related proteins from sporophyte (n = 5), male (n = 5) and female (n = 5) tips were constructed utilising average TPM as input data. Bar graphs showing relative gene expression of rhodophyte collagen-alpha-like proteins (RCAPs; putative secreted proteins) identified in this study were constructed utilising log2(TPM+1) with error bars representing mean ± standard deviation.

## 3. Results and Discussion

A comprehensive overview of expressed proteins in *A. taxiformis* was established for two different life stages (gametophyte and sporophyte) using isolation methods for soluble and insoluble proteins. A total of 741 and 2007 unique non-redundant proteins were identified using genome-derived and transcriptome-derived databases, respectively (Figure 2, inset and Appendix A). The comparative number of proteins identified from the two life stages and extraction method was further investigated (Figure 2). Overall, the highest number of proteins was identified from the female gametophytes, while the urea extraction provided the greatest overall yield of unique proteins. Upon comparison of *A. taxiformis* reference protein databases used for protein identification (genome and transcriptome), 54% of proteins were common in both (based on *e*-value cut-off 8.79 × 10^−5^). Although the transcriptome-derived protein database identified more proteins, a large proportion of proteins matched to non-seaweed species (Appendix A), which was likely a result of epiphyte and endophyte microorganisms on the field-collected samples. In support of this, the Blast top-hit species indicated that many proteins were from fungi species. Fungal endophytes colonise the inner part of tissues, and although they are thought to play an important role in the production of secondary metabolites, not much is known from seaweeds [25].

The large number of diverse proteins we observed compared favourably against other red seaweed proteomic studies [16,26]. In our study, the utilisation of morphologically distinct life stages (filamentous sporophyte and foliose gametophyte) may have contributed to the relatively large numbers of proteins identified. If we had only considered any single life stage, then we would have reported a protein number of between 281 (male) to 421 (sporophyte) from the genome database. This may also be explained by differences in extraction and/or identification procedures, as other red seaweed proteomic studies have employed extraction followed by gel-based fractionation (1- or 2-dimensional). For instance, a phenol/chloroform extraction of the red algae *G. changii*, followed by 2-D electrophoresis, resulted in the elucidation of 15 proteins [16]. In addition, using phenol-based extraction protocols, high-molecular-weight proteins in 1-D electrophoresis were elucidated from *E. cottonii* (presently *Kappaphycus alvarezii*), however, the total number of identified proteins was not reported [17]. In the red seaweed *Eucheuma denticulatum*, a hot water extraction method yielded 66 different proteins that were largely extracellular, which as a relatively simple procedure could be easily up scaled for industry [26]. However, if the downstream purpose was to assess bioactivity, the hot water extraction, as well as gel-based fractionation approaches, would not be suitable.

### 3.1. Gene Ontology Analysis

Given our focus on seaweed-specific proteins, we further investigated the proteins in the genome-derived protein database to determine how unique they are. We identified 4152 unique peptides using the genome-derived database (Appendix A). Identified *A. taxiformis* proteins were assigned to biological process, molecular function and cellular component categories using Gene Ontology (GO) analysis (Appendix A). Of the 741 proteins, 92.04% were successfully annotated, while the remainder had no BLAST matches. The distribution of the top 10 GO terms (Level 3) for the three categories is shown in Figure 3A. According to the GO subcategories, the most abundant biological process terms were cellular metabolic process, organic substance metabolic process and primary metabolic process. For molecular function, the most abundant GO terms were heterocyclic compound binding, organic cyclic compound binding and ion binding. Further annotation of proteins assigned to the molecular function, at Level 7, supported a broad distribution of ion-binding proteins (iron, zinc, manganese, potassium and copper ion-binding proteins) (Figure 3B). Biosorption is a low-cost physiochemical process involved in the binding of metal ions using natural biosorbents, based on a variety of mechanisms (e.g., adsorption, absorption, ion exchange, surface, complex formation, chelation, precipitation, bioaccumulation inside cells and binding to proteins and other intracellular components) [27,28]. Seaweeds have intrinsic biosorbent properties and exhibit a high metal ion binding capacity within contaminated water, primarily attributed to the various cell wall functional groups [29]. Possibly the most well studied seaweed in this area is the red seaweed, *Gracilaria verrucosa*, which has high biosorption capacity for the removal of nickel, and therefore is considered as a promising cost-effective biosorbent for wastewater treatment [28]. Herein, our extraction and identification of various ion-binding proteins in *A. taxiformis* suggested it has the capacity to be an effective natural biosorbent in similar clean-up biotechnology, as either a sorbent or through bioaccumulation.

For the cellular component category, the major GO terms were intracellular anatomical structure, organelle and cytoplasm. We observed that out of 126 unique proteins identified that were cytoplasmic, 50% were exclusive to urea extracts, while relatively few (12) were exclusive to Tris-HCl extracts (and 53 were shared). Similarly, of the total membrane-related proteins identified (99), the majority were exclusive to urea extracts (51), while 21 were exclusive to Tris-HCl extracts (and 27 were shared). This outcome may be attributed to urea being a chaotropic agent that disrupts hydrogen bonds and hydrophobic interactions, both between and within proteins, allowing them to be removed from membranes [30]. However, since our study was not quantitative, the outcome does not reflect protein abundance.

### 3.2. Comparison between Life History Stage and Sex

Of the 741 genome-derived proteins identified, 87 were common among all life history stages, while 210, 163 and 140 were uniquely identified in sporophytes and male and female gametophytes, respectively (Figure 4A). Quantitative RNA-seq gene expression analysis of the corresponding proteins demonstrated variation between life stage and sex, which helps support the differential protein identification observed (Figure 4B and Appendix A). In sporophytes, those genes exclusive or relatively highly expressed were generally classified into the functional categories of enzyme and structural, as well as hypothetical/unknown and DNA/RNA-related (Figure 4C). The sporophyte proteome yielded 41 proteins related to structure (e.g., actin, cell surface glycoprotein 1-like, tubulin alpha-1/2/3 chain, tubulin beta-1 chain and endomembrane protein-like protein), which had relatively low gene expression in gametophytes. We postulate that due to the sporophyte having a more delicate structure, consisting of filaments of cells [31], the structural components within these may be relatively easy to isolate. Proteins identified and relatively highly expressed in males were generally classified into the categories of hypothetical/unknown, enzyme and DNA/RNA-related, while in females there was a relative abundance of enzymes, hypothetical/unknown and photosynthesis-related (27 exclusive to female) (Figure 4C).

### 3.3. Targeted Identification of Photosynthesis-Related and Stress-Related Proteins

Like all phototrophs, productivity in seaweeds is driven by the absorption of light to convert into chemical energy, where phycobilisomes serve as the primary light-harvesting antenna for the photosystem II (PSII) complex [32]. This proteomic study allowed for the identification of 44 photosynthesis-related proteins, including phycobilisomes, photosystem I, photosystem II and ATPase (Appendix A). The phycobilisome protein complex is attached to the outer side of the inner thylakoid membrane and comprises stacked hexameric and trimeric disks of three types of proteins (i.e., phycoerythrin, phycocyanin and allophycocyanin), which differ in protein identity, chromophore type, attachment and their relative location in the phycobilisome complex [33]. Phycobiliproteins function as primary light absorbers with greater adaptability and light-capturing capacity, hence possessing higher photosynthetic capacity compared to chlorophyll and carotenoids [32]. In red seaweeds, most of our knowledge on photosynthesis-related components is from *Kappaphycus alvarezii*, where transcriptome shifts in response to CO_2_ enrichment detected 9 up-regulated genes, including 4 photosystem I (PSI) and 5 PSII proteins, out of 38 candidate photosystem proteins [34].

Interestingly, our gene expression analysis demonstrated higher expression in the female samples for most of the light-harvesting proteins identified, encompassing phycobilisome linker proteins and R-phycoerythrin (Figure 5A). This reflects our exclusive identification of photosynthesis-related proteins in female gametophytes (see Figure 4C). Similar observations from another red seaweed, *Pyropia *haitanensis**, report that gametophytes contain significantly higher contents of phycoerythrin, allophycocyanin and chlorophyll α compared to sporophytes [35]. We speculate that the higher expression of photosynthesis-related proteins in females are directed towards related energy requirements that females use for reproduction [36].

*Asparagopsis taxiformis* is usually found attached to a solid substrate in shallow subtidal areas, and therefore encounters challenging environmental conditions [37]. Abiotic stresses, including light, salinity, temperature fluctuations, nutrient availability and desiccation, confer serious damage on various molecular components, in particular the photosynthetic machinery [38]. Targeted seaweed proteomics, particularly in response to environmental changes, has gained significant popularity as a method to detect relevant proteins. Previous studies showed inhibition of photosynthesis and energy metabolisms in the course of desiccation stress in *Pyropia haitanensis* [39] and *Pyropia orbicularis* (previously known as *Pyropia columbina*) [40].

Heat shock proteins (HSPs) are a large family of conserved molecular chaperones that are renowned for their roles in protein maturation, re-folding and degradation in the course of stress-induced response [41]. HSPs have been classified into five families based on molecular size in plants and other organisms, HSP100, HSP90, HSP70, HSP60 and small HSPs (sHSPs/HSP20), which can be induced by heat and other related stresses [42]. Following artificial infection of *Pyropia yezoensis* with a *Pythium porphyrae* spore, HSP20 was reported as significantly up-regulated [43].

In this study, we identified 58 stress-related proteins in *A. taxiformis*, including 8 HSPs and 5 vanadium-dependent haloperoxidases (Appendix A). There was varied gene expression of stress-related proteins across male, female and sporophyte samples (Figure 5B). However, it is worth mentioning that of the identified HSPs, two (Ata01778 and Ata09909) were much more highly expressed in male and female than sporophyte samples. Similar gene expression profiles were observed for sHSPs in *Pyropia yezoensis* based on transcriptomics, which suggested a requirement to maintain metabolism and cellular protection during alteration of gametophyte to sporophyte in response to high-temperature stress [42].

### 3.4. Targeted Identification of Predicted Secreted Proteins

The *A. taxiformis* genome predicted 345 genes that encode secreted proteins [22], of which 40 were detected in our proteomic analysis (Appendix A). This included 10 rhodophyte collagen-alpha-like proteins (RCAPs), as well as protein disulphide isomerases (PDIs), vanadium-dependent haloperoxidases (VHPOs), an animal heme-peroxidase homologue, and proteins that annotated as hypothetical/uncharacterised/novel. The RCAPs are presumed to be involved in bioadhesion as a secreted sticky mucilage that facilitates substrate attachment [22]. In support of this, we observed a relatively higher gene expression for most of the identified RCAPs in sporophyte samples compared to male and female tips, particularly for Ata10851, Ata10856, Ata06706 and Ata09078 (Figure 6); this may reflect the requirement for substrate attachment in sporophytes. However, other genes were relatively high in gametophytes (e.g., Ata02400, Ata10426 and Ata02738), which indicates these may have alternate functions.

In eukaryotes, PDIs are a protein family that enable oxidative folding of newly synthesised proteins by catalysing the formation, breakage and rearrangement of disulphide bonds in newly synthesised secretory proteins [44]. Disulphide bonds play a fundamental role in stabilising protein structures, which ensure their performance of normal biological functions. This occurs en route through the secretory pathway, within the endoplasmic reticulum [44]. In plants, there are six diverse subfamilies of PDIs, however, little is known in seaweeds. Research has recognised the importance of PDIs in the stress response, in addition to endoplasmic reticulum stress, which is related to unfolded or misfolded proteins in *Arabidopsis thaliana* [45,46]. PDI expression was reported up-regulated in terms of transcript level and enzyme activity by both high-intensity illumination (1200 µmol photons. M^−2^. S^−1^) and hypersalinity (90) [47]. Three proteins were identified as PDIs in the *A. taxiformis* proteome (*e*-value = 0), of which two (Ata01812 and Ata03103) were isolated from sporophytes (using both urea and Tris-HCl extraction) and female samples (using only urea extraction), while one (Ata07843) was isolated from all the samples using both extraction methods.

The VHPOs are enzymes involved in the biosynthesis of halogenated natural products [48], conventionally classified as chloro-, bromo- and iodo-peroxidases [49]. Interestingly, VHPOs play a role in early embryogenesis of brown algae, as well as contributing to the metabolism of reactive oxygen species (ROS) [50]. In this study, we identified three VHPOs (Ata01934, Ata07479 and Ata10244) that are predicted to be secreted. More specifically, all three VHPOs were isolated from sporophytes using urea extraction, which is assumed to be involved in seaweed defence and stress response. However, the specific site of their function is still unknown.

## 4. Conclusions

This study presents the first comprehensive proteome for the red seaweed *Asparagopsis taxiformis* (Lineage 6). We also highlighted the significance of using multiple protein extraction methods to enhance proteome coverage. Future use of this reference proteome will see the exploration of proteins associated with the differentiation between gametophyte and sporophyte, including alternation of generations in the life history. The underlying molecular mechanisms of reproduction would shed light on the key molecular components for the triphasic life cycle in this seaweed. In addition, expanding on quantitative protein analysis and integrating that with gene expression at different life stages, or more specifically between male and female gametophytes, would establish a systems approach that could enable a better understanding of changes that occur. This would fundamentally contribute to a better understanding of various basic concepts of growth, metabolism, biosynthesis and reproduction that is relevant to scaling up *A. taxiformis* aquaculture. Furthermore, future studies focusing on *Asparagopsis* ion-binding proteins could lead to approaches for heavy metal bioremediation relevant to clean-up biotechnology, such as the treatment of industrial wastewaters at a large-scale. This could be an added benefit for any coastal farming of seaweed if concentrations do not impact the application of the seaweed as a livestock feed additive. There is also potential for the extraction of bioactives followed by the use of the biomass residue with polysaccharides and proteins as biosorbents.

## Figures and Tables

**Figure 1 biology-12-00167-f001:**
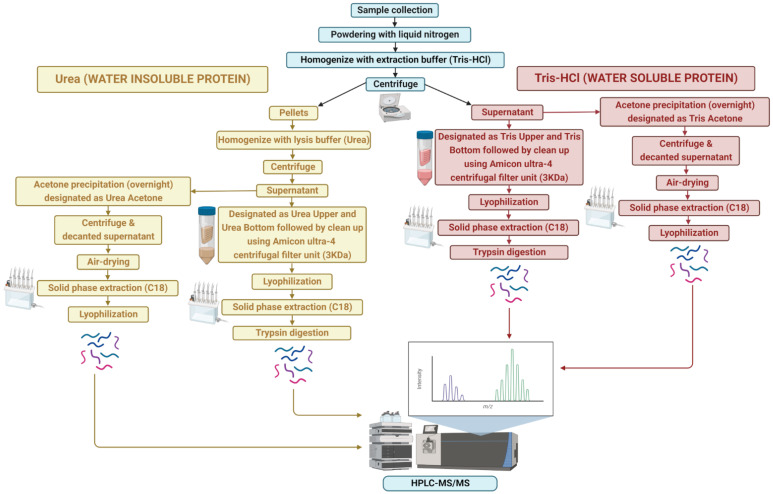
Overview of sample extraction and protein identification workflow for water-soluble (Tris-HCl) and -insoluble (urea) proteins.

**Figure 2 biology-12-00167-f002:**
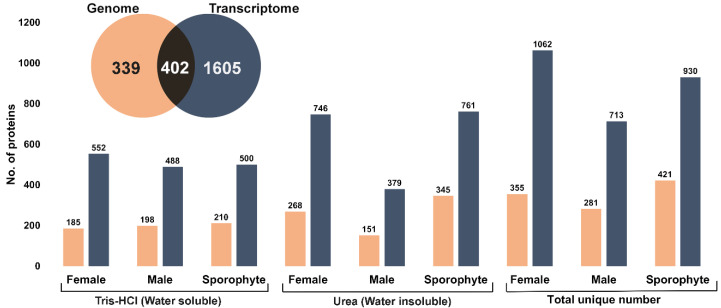
Summary of total proteins identified from female, male and sporophytes using two different extraction methods and two protein databases (genome and transcriptome). Inset: Venn diagram shows the total unique number of proteins identified from genome- and transcriptome-derived protein database.

**Figure 3 biology-12-00167-f003:**
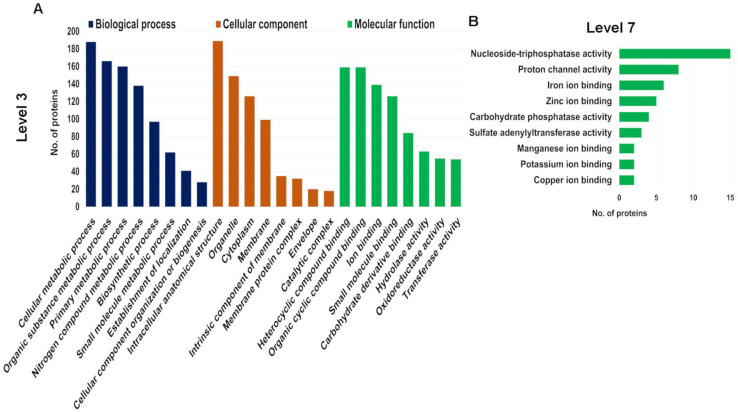
Gene Ontology (GO) analysis of all identified proteins in genome-derived protein database within biological process, molecular function and cellular component. (**A**) Top 8 enriched GO terms at Level 3. (**B**) Top 9 enriched GO terms for molecular function at Level 7.

**Figure 4 biology-12-00167-f004:**
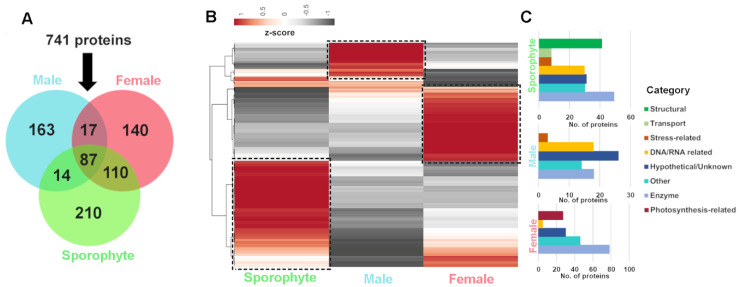
Summary of *A. taxiformis* proteomic identification and corresponding gene expression in female, male and sporophyte. (**A**) Venn diagram showing the number of common and uniquely extracted proteins identified from sporophyte, male and female tips. (**B**) Heatmap showing relative gene expression (using Z-score) for proteins isolated from sporophyte, male and female tips. (**C**) Bar graphs showing the stage-biased protein functional categories for sporophyte, male and female tips. ‘Boxed’ areas in B further classified into functional categories, as shown in C. See Appendix A for gene information.

**Figure 5 biology-12-00167-f005:**
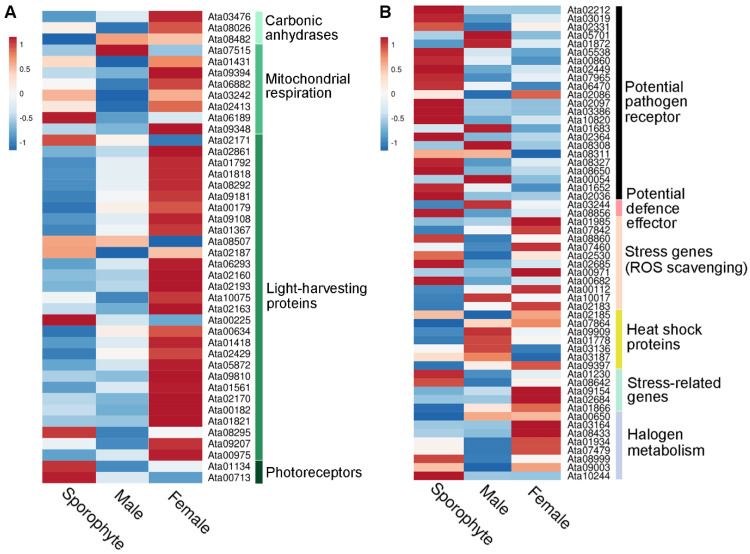
Overview of relative gene expression of identified photosynthesis-related and stress-related proteins from sporophyte, male and female tips. (**A**) Heatmap showing relative gene expression (using average TPM) for photosynthesis-related proteins isolated from sporophyte, male and female tips. (**B**) Heatmap showing relative gene expression (using average TPM) for stress-related proteins isolated from sporophyte, male and female tips. See Appendix A for gene information.

**Figure 6 biology-12-00167-f006:**
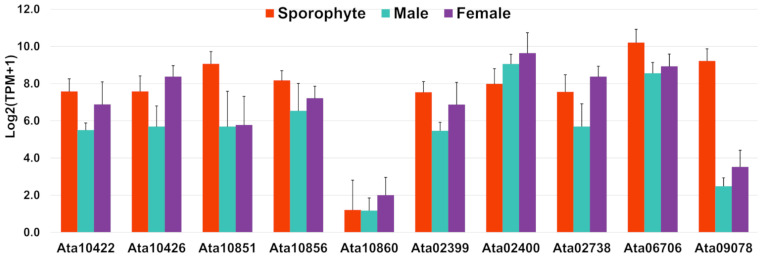
Bar graphs showing relative gene expression with log2(TPM+1) of rhodophyte collagen-alpha-like proteins, RCAPs (putative secreted proteins) identified in this study. TPM represents transcripts per million and error bars represent mean ± standard deviation.

## Data Availability

Raw data of all transcriptome sequencing were submitted to the NCBI Bioproject and SRA database under a Bioproject accession number PRJNA869876. Mass spectrometry peptide raw data were deposit into the PRIDE database under accession number PXD035669.

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
