# Peer review of "A Proteomic Analysis for the Red Seaweed Asparagopsis taxiformis"

_biology, 2023, doi:10.3390/biology12020167_

Round 1

Reviewer 1 Report

The manuscript deals with proteomic and transcriptomic analysis of Asparagopsis taxiformis using two life cycle stages. This study can complement the information existent in red algae.

The methodology was according to the objective of the work, and the experimental details are enough. Also, the data analyses are robust.

However, there are some suggestions for improve the information.

Change the title, this title is not adequate to the finds of this work.

Line 41, delete marine macroalgae, is redundant.

Line 41-42, insert references

Line 48, insert the authorships

Figure 1, change the seaweed image, because don’t correspond to the species.

Excellent description of the methodology, many details, congratulations.

Figure 3 , low quality, please change

Major discussion is required about the results of ion binding proteins, are there clear difference between stage? Please incorporate a life cycle of the species as supplementary material

Check all the manuscript, there are words in capital letters, as words in bold or in cursive when are not necessary

Please, check all the seaweed taxonomy (algaebase.org), for example, you must change Pyropia columbina to Pyropia orbicularis (line 353) (see  https://www.biotaxa.org/Phytotaxa/article/view/phytotaxa.158.2.2)

Major information is required about the ecological context of the findings of the work. Are there ecological importance between stages?-

Author Response

Reviewer 1

Comment 1. Change the title, this title is not adequate to the finds of this work.

Response: The title has now changed fromA proteotranscriptomic investigation of two life history stages for the red seaweed Asparagopsis taxiformis” to “A proteomic analysis for the red seaweed Asparagopsis taxiformis”.

Comment 2. Line 41, delete marine macroalgae, is redundant.

Response: As suggested, marine microalgae has now been removed from line 41.

Comment 3. Line 41-42, insert references

Response: As suggested, we have included authorship in Line 48.

Comment 4. Line 48, insert the authorships

Response: As suggested, we have included authorship in Line 48.

Comment 5. Figure 1, change the seaweed image, because don’t correspond to the species.

Response: We acknowledge this, and have now removed the seaweed image from the Figure 1.

Comment 6. Excellent description of the methodology, many details, congratulations.

Response: Thank you and we appreciate this comment.

Comment 7. Figure 3, low quality, please change

Response: We have now modified Figure 3 to be of higher quality, as suggested.

Comment 8. Major discussion is required about the results of ion binding proteins, are there clear difference between stage? Please incorporate a life cycle of the species as supplementary material.

Response: From Gene Ontology analysis, we couldn’t find any clear difference of ion binding proteins between stages. Therefore, it was not noted in the more focused results and discussion.

Comment 9. Check all the manuscript, there are words in capital letters, as words in bold or in cursive when are not necessary

Response: Thank you, and we have revised the manuscript.

Comment 10. Please, check all the seaweed taxonomy (algaebase.org), for example, you must change Pyropia columbina to Pyropia orbicularis (line 353) (see  https://www.biotaxa.org/Phytotaxa/article/view/phytotaxa.158.2.2)

Response: Agreed, and this has now been changed.

Comment 11. Major information is required about the ecological context of the findings of the work. Are there ecological importance between stages?

Response: Given that the stages (gametophytes and sporophyte) used in this study were the same species, Asparagopsis taxiformis, we could not pinpoint any ecological importance/differences between stages.

Reviewer 2 Report

Zubaida et al. reported the overview of expressed proteins in A. taxiformis using proteotranscriptomic techniques. Using solube and insoluble extraction methods, they improved the coverage of identified proteins, which would be benefit for the study of  A. taxiformis. In my opinion, the research is interesting. However, I still have several comments.

1. Introduction
Acturally, in the present study, the authors combined the proteomics and transcriptomics study. However, they didn't refer the significance of transcriptomics study. The second paragraph showed the powerful of proteomics. It seems the transcriptomics data is just used for the protein identification. Right?
Usually, for those species with known genome, we used the genome for the protein identification in proteomics study. Transcriptomics is not necessary. So, please clarify the role of trancriptomics study and the advantage of proteotranscriptomic study.

2. M & M
The authors should clarify the criteria of identified proteins for example how many unique peptides were identified for them. What kinds of statistical analysis method were used in the present study?

3. Results
Based on the fig 2, it seems when using transcriptomic-derived databases, you got the more proteins than genome-derived databases. However, why in fig4, we use the identified proteins from genome-derived databases rather than transcriptomic-derived databases?

In the section 3.3, the PS-related and stress-related proteins were from which results, transcriptomic or genome-derived database?

In fig6, the error bars is so large that we cann't know whether the gene expressions are significance among female, male and sporophyte. The statistical analysis is needed.

In summary, the authors have done many works, including the multiple protein extraction methods to enhance proteome coverage. However, the authors focused on the PS and stress related proteins (3.3) and secreted proteins (3.4) discussion, I didn't know which results (fig or table) guide us to notice PS, stress and secreted proteins.

Author Response

Comment 1. Introduction: Actually, in the present study, the authors combined the proteomics and transcriptomics study. However, they didn't refer the significance of transcriptomics study. The second paragraph showed the powerful of proteomics. It seems the transcriptomics data is just used for the protein identification. Right? Usually, for those species with known genome, we used the genome for the protein identification in proteomics study. Transcriptomics is not necessary. So, please clarify the role of trancriptomics study and the advantage of proteotranscriptomic study.

Response: We agree and now the Introduction has been updated with the importance and benefit of integrating transcriptome knowledge with proteins identified (line 59-67).

Comment 2. M & M: The authors should clarify the criteria of identified proteins for example how many unique peptides were identified for them. What kinds of statistical analysis method were used in the present study?

Response: We have now integrated the number of identified unique peptides into the result section, as suggested (line 237-238) and included in File S3 (line 401-403).

De novo sequencing of proteins, database searches and characterisation of specific post-translational modifications were used to analyse the raw data; false discovery rate was set to ≤ 1%, and [-10*log(p)] was calculated accordingly, where p is the probability that an observed match is a random event (line 175-178).

Z-score was used for heatmap construction (line 193-195). In addition, we updated the e-value cut-off 10-3 which was used for protein functional annotation (line 184-185).

Comment 3. Results: Based on the fig 2, it seems when using transcriptomic-derived databases, you got the more proteins than genome-derived databases. However, why in fig4, we use the identified proteins from genome-derived databases rather than transcriptomic-derived databases?

Response: Given our focus on seaweed-specific proteins, we further investigated the proteins in the genome-derived protein database to determine how unique they are, line 233-234. From 3.1 Gene ontology analysis sections onwards, all the results were obtained from genome-derived database.

In the section 3.3, the PS-related and stress-related proteins were from which results, transcriptomic or genome-derived database?

Response: In the section 3.3, the PS-related and stress-related proteins were obtained from the genome-derived database.

In fig 6, the error bars is so large that we cann't know whether the gene expressions are significance among female, male and sporophyte. The statistical analysis is needed.

Response: Figure 6 has now been updated and changed with Log2 TPM (line 353-355), as suggested by Reviewer 3 as well.

In summary, the authors have done many works, including the multiple protein extraction methods to enhance proteome coverage. However, the authors focused on the PS and stress related proteins (3.3) and secreted proteins (3.4) discussion, I didn't know which results (fig or table) guide us to notice PS, stress and secreted proteins.

Response: Figure 5 (line 309-314) and File S5, Tab 1&2 (line 400-405), represented PS and stress related proteins (3.3) and Figure 6 (line 348-351) and File S5, Tab 3 (line 406-408) represented secreted proteins (3.4) discussion.

Reviewer 3 Report

In the manuscript authors are trying to elucidate underlying difference of life stages of seaweed Asparagopsis taxiformis by exploring proteomics and gene expression. They have tried multiple protein extraction strategies to explore the different stages of A.taxiformis, and their effort has made robust information on unexplored proteins, pathways and expression. 
  Following are the few comments.  
  1. The authors have done lots of global analysis on both protein and gene expressions but having some validation for protein expression in different life stages or more specifically between male and female A.taxiformis would strengthen their data.(figure 4)
  2. They have done RNA seq-analysis, but they have limited the analysis to gene expression levels,  it would be worth file to do some IPA analysis and also validate gene expression levels.(figure 5)
  3. in figure 6 they have shown TPM for comparison of gene expression, this should be a very cautious observation, TPM may not be absolute measure, different library prep strategies have different biasness.
  The manuscript should be thoroughly revised for some spell checks and grammar mistakes.

Author Response

Comment 1. The authors have done lots of global analysis on both protein and gene expressions but having some validation for protein expression in different life stages or more specifically between male and female A.taxiformis would strengthen their data.(figure 4).

Response: This study was focused on qualitative analysis (line 268-269) and, therefore, we could not accurately infer protein expression with male or female. However, the integrated gene expression data suggests that quantitative analysis should be investigated in the future. This idea has now been included in the Conclusion as future direction section, line 379-382.

Comment 2. They have done RNA seq-analysis, but they have limited the analysis to gene expression levels, it would be worth file to do some IPA analysis and also validate gene expression levels. (figure 5)

Response: We acknowledge this, and have now included this idea in the Conclusion as future directions section, line 379-382.

Comment 3. In figure 6 they have shown TPM for comparison of gene expression, this should be a very cautious observation, TPM may not be absolute measure, different library prep strategies have different biasness.

Response: Agreed and Figure 6 has now been changed to use Log2 TPM (line 353-355), as suggested. We highlight that further research will focus on these genes/proteins, that will include support for gene expression.

Comment 4. The manuscript should be thoroughly revised for some spell checks and grammar mistakes.

Response: Thank you, and we have revised the manuscript.